# Lived experiences of people living with HIV—A qualitative exploration on the manifestation, drivers, and effects of internalized HIV stigma within the Malawian context

**Moses Kelly Kumwenda**[1,2]*, **David Black Kamkwamba**[3], **Maureen Leah Chirwa**[3], **Kasoka Kasoka**[4], **Miriam Taegtmeyer**[5], **Tessa Oraro-Lawrence**[4], **Lucy Stackpool-Moore**[6]

1 Gender in Health Associate Group, Malawi Liverpool Wellcome Programme, Blantyre, Malawi, 2 Helse Nord TB Initiative, Kamuzu University of Health Sciences, Blantyre, Malawi, 3 Prime Health Consulting and Services, Lilongwe, Malawi, 4 International AIDS Society, Geneva, Switzerland, 5 Liverpool School of Tropical Medicine, Liverpool, United Kingdom, 6 Watipa, London, Australia

* kumwenda@gmail.com

## Abstract

### Introduction

HIV-related internalized stigma remains a major contributor to challenges experienced when accessing and providing HIV diagnosis, care and treatment services. It is a key barrier to effective prevention, treatment and care programs. This study investigated experiences of internalized stigma among people living with HIV in Malawi.

### Methodology

A participatory cross-sectional study design of participants from eight districts across the three administrative regions of Malawi. Data were collected using Key Informant Interviews (n = 22), Focus Group Discussions (n = 4) and life-stories (n = 10). NVIVO 12 software was used for coding applying both deductive and inductive techniques. Health Stigma and Discrimination Framework was used as a theoretical and analytical framework during data analysis.

### Results

Overt forms of stigma and discrimination were more recognizable to people living with HIV while latent forms, including internalized stigma, remained less identifiable and with limited approaches for mitigation. In this context, manifest forms of HIV-related stigma intersected with latent forms of stigma as people living with HIV often experienced both forms of stigma concurrently. The youths, HIV mixed-status couples and individuals newly initiated on ART were more susceptible to internalized stigma due to their lack of coping mechanism, unavailability of mitigation structures, and lack of information. Broadly, people living with HIV found it difficult to identify and describe internalized stigma and this affected their ability to recognize it and determine an appropriate course of action to deal with it.

**Data Availability Statement:** The minimal data set underlying the results described in this manuscript

can be found at the Qualitative Data Repository (https://doi.org/10.5064/F6UJUEP3.

**Funding:** KK. LS-M and TO-L received funding through the International AIDS Society (IAS) from the Bill and Melinda Gates Foundation (https://www.gatesfoundation.org/). The funders had no role in study design, data collection and analysis, decision to publish, or preparation of the manuscript.

**Competing interests:** The authors have declared that no competing interests exist.

## Conclusion

Understanding the experiences of internalized stigma is key to developing targeted and context specific innovative solutions to this health problem.

## Introduction

HIV stigma remains pervasive at each point of the HIV care continuum and is profoundly ingrained within social, economic, cultural and historical contexts [1,2]. The manifestations of HIV stigma are socially constructed and characterize some people or groups of people as discreditable or unworthy [3,4]. A recent systematic review and meta-analysis reported significant associations between reporting stigma including internalized stigma and having CD4 counts of less than 200, reduced HIV medication adherence, overall poor access to care, and length of time since diagnosis [5]. Globally, the 2020 UNAIDS Global AIDS Update observed that HIV stigma persist to obstruct people living with HIV from making use of HIV-related services and that discriminatory attitudes towards people living with HIV continue to be prevalent in many countries with the available legal systems unable to provide protection [6–8].

Stigma that is rooted in the self, also referred to as internalized stigma, is quite predominant but difficult to identify and extremely hard to intervene [9]. Internalized stigma has been defined as "a subjective process, embedded within a socio-cultural context, which may be characterized by negative feelings (about self), maladaptive behaviour, identity transformation, or stereotype endorsement resulting from an individual's experiences, perceptions, or anticipation of negative social reactions on the basis of their [socially devalued identity or] illness" [10]. In the context of HIV, self-stigma surfaces when an individual internalise negative attitudes and experiences such as shame, blame, hopelessness, guilt and fear of early death and negative social experiences associated with being HIV positive [11]. Internalized HIV stigma becomes an issue of concern when a person living with HIV begins to conform to stigmatizing stereotypes attached to HIV.

The internalization of stigma have been reported to affect quality of life and health outcomes including mental health issues such as anxiety and depression [12,13], access to HIV-related services [14], adherence to antiretroviral treatment (ART) [15] and willingness to disclose [16]. In South Africa, people living with HIV who internalize stigma were reported to engage in avoidant coping as a resilience mechanism which in turn negatively affects treatment and health outcomes [17].

In Malawi, HIV stigma remains a significant challenge even after the roll out of effective treatment and prevention programmes. About a quarter of people living with HIV report having been physically assaulted, with social exclusion being more pronounced in this group [18,19]. The main sources of HIV stigma among people living with HIV in Malawi include distant relatives, friends and church members because of its association with behaviours that are often judged such as prostitution or promiscuity [20]. While acknowledging complexities in determining and identifying root causes of HIV stigma, examples of stigma sources as spelled by Hammarlund et al (2018) include social institutions, public opinion and self [21–23].

HIV stigma negatively impacts Malawian women more than men due to contextual social and economic inequalities that make women living with HIV to become more vulnerable to emotional and physical abuse perpetrated on the basis of their HIV status [24,25]. Nevertheless, HIV stigma in this context also intersects with masculinities over the life course of men but does not translate to poorer health outcomes for women living with HIV [26,27]. The

expression of HIV stigma has been fluid since the genesis of HIV epidemic, altering its manifestations and characteristics as HIV prevention, treatment and care programming have been developing and evolving over time [28].

Despite the growing debate on getting to the heart of HIV stigma [21,29], researchers have traditionally focused on overt forms of stigma [30,31]. Recently, there have been a growing attention given to internalized forms of stigma that are subtle and operate under the surface [13]. This study sought to conduct in-depth analyses of the country contexts and latest national progress, policies, and research to contextualize, triangulate, interrogate, validate, and enhance the evidence base in the context of lived realities in Malawi about HIV-related self-stigma. Here, we report on internalized stigma lived experiences of people living with HIV within the Malawi context. We focus on the manifestation, drivers and effects of internalized HIV stigma. Understanding contextual lived experiences of internalized HIV stigma is a first step to broadly determine befitting mitigation options within resource constrained settings.

## Methods

### Theoretical orientation

A grounded theory methodological orientation [32] was used in this study and underpinned by the Health, Stigma and Discrimination Framework to understand internalized HIV stigma [33]. This theory was purposively selected based on the research purpose to learn about lived experiences of internalized stigma among people living with HIV while making visible situations, relationships, and hierarchies of power. In addition to using the Health, Stigma and Discrimination Framework to guide our data analysis, this framework also informed the research design, and the development of our data collection instruments. The framework was chosen because it articulate the stigmatization process as it unfolds across the socio-ecological spectrum within the context of health, varying across economic contexts [33] and helps to navigate and analyze drivers and facilitators; stigma manifestations that influence a range of outcomes among the affected individuals, as well as organizations and institutions, that ultimately impact health and society (see Fig 1).

### Design and study setting

This qualitative study employed a participatory cross-sectional design involving Key Informant Interviews (KIIs), Focus Group Discussions (FGDs) and life stories, to gain in-depth insights on vulnerable groups and people living with HIV experiences and the implications of these on access to HIV services. The mix of the different groups of study participants chosen for KIIs, FGDs and life stories made qualitative data very rich and insightful to address research questions.

Key informant interviews were used to obtain in-depth and contextual information about internalized stigma broadly from individuals that had vast knowledge about this subject matter. These interviews were conducted with national level participants namely representatives from the Department of HIV and AIDS; district level participants (i.e., District HIV and AIDS Coordinators, District Social Welfare Officers, representatives from NGOs working on HIV and AIDS related projects); and community level participants (i.e., people living with HIV, traditional and religious leaders and Health Care Workers). These participants were included in the study based on their knowledge of the HIV programming in Malawi, including the established mechanisms for mitigating (internalized) HIV stigma in communities.

Focus group discussions (FGDs) were chosen because of their ability to use group interaction to generate data on collective experiences about internalized HIV stigma. Group discussions were conducted at community level with village health teams from CBOs, (PWD, female

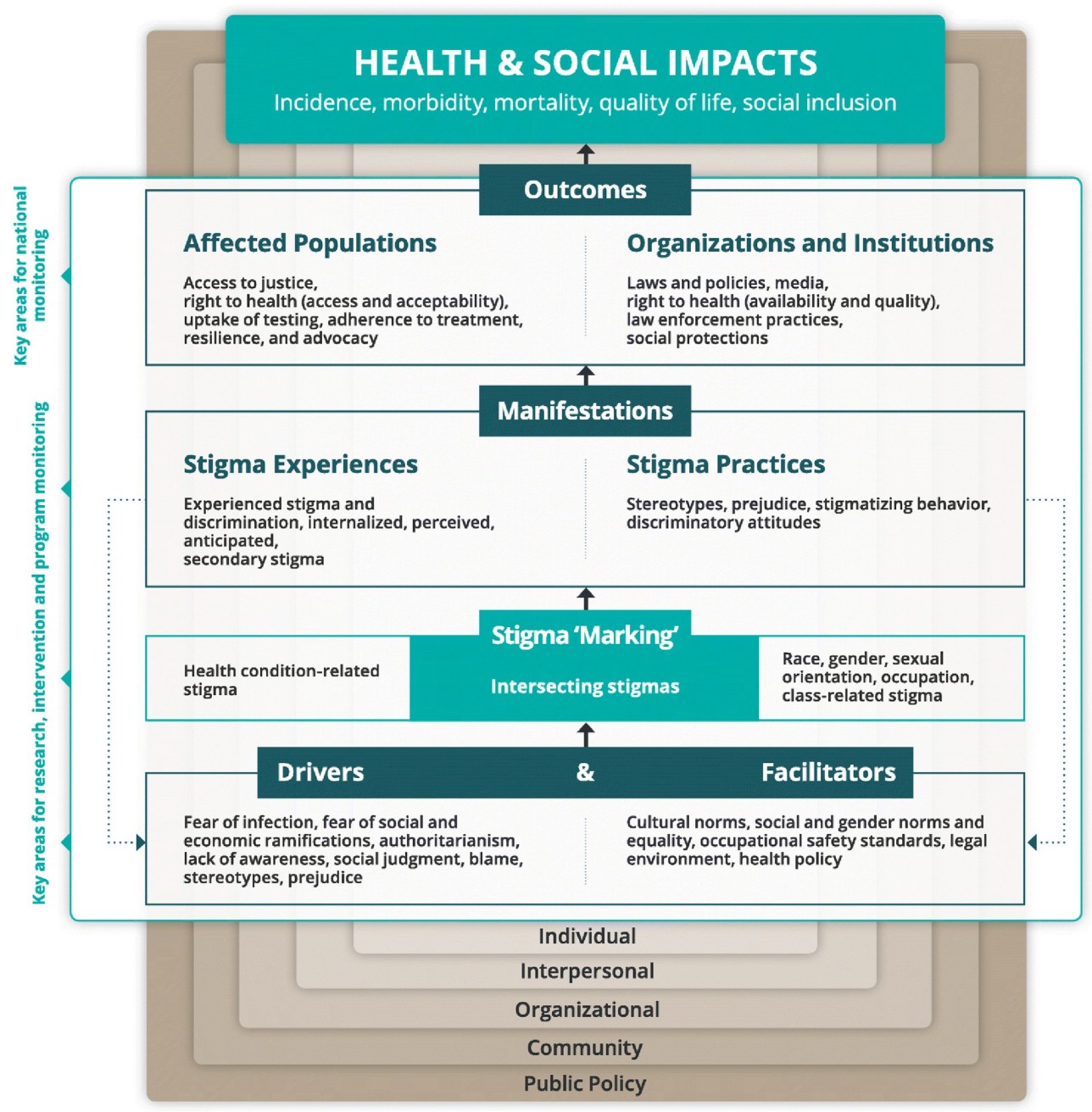

**Fig 1. Health stigma and discrimination framework.**

sex workers and people living with HIV). Efforts to include representatives from male sex worker communities were unsuccessful as two individuals that were eligible and approached for a group discussion failed to present. Adolescent girls and boys participated in these FGDs but through separate discussions.

A life story methodology was also employed to capture lived experiences of people living with HIV with reference to internalized HIV stigma. Once an inspirational story was identified in KIIs and FGDs for potential inclusion in life stories, repeat in-depth interview (IDI)

centering on the fascinating story was conducted by DBK an experienced qualitative researcher. Life-story study participants were drawn from FGDs participants to explore lived experiences emanating from FGDs that were difficult to follow in a group discussion context.

Qualitative data was collected from adults (18 years and above) in eight districts across the four regions of Malawi: Rumphi (Northern region), Ntcheu and Lilongwe (Central region), Machinga and Zomba (Eastern region) and Mulanje, Mwanza, Neno and Thyolo (Southern region). Non-probabilistic purposive sampling was used to identify study districts and select study participants who represented a spectrum of people living with HIV, Key Populations (KPs), persons with disabilities (PDs), traditional leaders, members of community-based organizations (CBOs) and several government officials including policy makers, magistrates, social workers, police and healthcare workers. A heterogeneous sample of districts and study participants permitted the study to obtain a broad range of perspectives and triangulation of views from multiple data sources in order to enhance truthfulness and completeness of data.

## Participants

Twenty-two KIIs, 5 FGDs and 10 life stories were done for this study (see Table 1) between 6[th] and 12[th] April 2021. Data collection for FGDs and KIIs continued until saturation of information was achieved. Districts were chosen depending on cultural diversity because this influences how stigma is experienced and understood. We included border districts because they form part of HIV hotspots in Malawi. More districts were included from the southern region because of the high HIV prevalence in that region.

## Participant recruitment and data collection

A nested sampling technique was used in this study. The initial participants for the KIIs and FGDs were purposively chosen while a sub-group of the life-story participants was purposively selected from the initial sample [34]. Participants for the initial sample were purposively selected based on their virtue of being people living with HIV, living with disability and HIV and being national/district/community stakeholders with roles and responsibilities in the development and implementation of the legal and policy instruments and guidelines. Additional participants eligibility criteria included being involved in the allocation of resources for assisting people living with HIV; healthcare providers involved in interventions designed to mitigate effects of HIV stigma; and those involved in fostering legal redress against HIV-

**Table 1. Participant demographic characteristics.**

| Data collection approach | Population | Sample size | Location |
|---|---|---|---|
| **Key Informant Interviews** | People Living with HIV | 5 | Rumphi, Mwanza, Neno, Thyolo |
| | Decision/policy makers | 5 | Rumphi, Machinga, Lilongwe |
| | Social service providers | 3 | Zomba, Mwanza, Mulanje |
| | Healthcare workers | 3 | Mwanza, Lilongwe, Ntcheu |
| | Traditional leaders | 2 | Mulanje, Ntcheu |
| | Young men and women | 2 | Zomba |
| | Transgender | 1 | Thyolo |
| | Person with disability | 1 | Thyolo |
| **Focus group discussions** | People Living with HIV | 1 session; n = 5 | Neno |
| | Female sex workers | 1 session; n = 12 | Mwanza |
| | People with Disabilities | 1 session; n = 4 | Thyolo |
| | MSM and TG | 1 session; n = 3 | Thyolo |

related human rights violations. People living with HIV, people with disabilities (PWDs) and KPs were recruited through researcher's existent collaboration and networks. Participants for the life stories were recruited based on interesting stories that emerged from data collected through KIIs and FGDs.

Data collection tools (KII and FGD guides) were developed by MKK, DBK and MLC (see Appendix 1). The development of the guides was framed by the study research questions which were aligned to the International AIDs Society's' (IAS) agenda on "Getting to the Heart of Stigma" and shaped by the Health, Stigma and Discrimination Framework. Data collection tools were then translated into local languages (Chichewa and Chitumbuka) and back into English; piloted to check their reliability in capturing data that addressed the research objectives. Members of our research teams included a mixture of individuals fluent in Chichewa and Chitumbuka languages. In terms of translation, HM translated guides from English to Chitumbuka while KN translated data collection tools from Chichewa to English. DBK back-translated and proof-read the Chichewa translation while MLC back-translated proofread the Chitumbuka translation.

Data collection guides were semi-structured and consisted of open-ended questions which were followed by probes that were based on participant's responses. Semi-structured interview guide contained questions on experiences of HIV stigma, local evidence on stigma reduction approaches, and inspirational stories on stigma reduction. Data collection was done physically (face-to-face) at a neutral venue for FGDs, and either workplaces or clinics for KIIs and life stories by a team of skilled qualitative research assistants. Interviews and group discussions were recorded using digital audio recorders and field notes. The KIIs and life stories lasted between 45 minutes and 1 hour while FGDs lasted between 45 minutes and 1 hour and 30 minutes (one-and-half hour). Non-participants were not available during the collection of all data. KIIs were conducted by experienced qualitative researchers DBK, KN, HM, KC, and TM who have vast skills in interviewing key informants. Group discussions were facilitated by HM, KC and DM who were well trained and had many years of collective experience in conducting FGDs. Focus group discussions facilitators were supported by TM and KN to take notes and managing logistics for the discussions. Life stories were collected by DBK and HM.

Data processing involved, assigning identifiers to audio recordings, transcribing data verbatim using Express ScribeTM, translating data into English. Transcribed data was checked for accuracy, comprehensiveness and intelligibility including clarity by MKK, DBK, and MLC. In this study, transcripts were not returned to participants for comments or correction.

## Analytical approach

Transcripts from KIIs and FGDs were uploaded into NVivo12 software for management and coding using a hybrid approach that combined deductive and inductive methods [35]. A codebook developed by MKK and DBK was based on domains directly extracted from the Health, Stigma and Discrimination Framework to serve as a priori codes [33]. Although, thematic data analysis were primarily done by MKK and DBK, review of results and discussion of the findings was undertaken by the entire research team and this included team collective decisions about the validity of the final themes and interpretation of findings [36]. Thematic data analysis commenced with reading and re-reading of data to familiarize with the data and to search for repeated patterns of meaning, common threads and issues of potential interest. Text was coded by MKK and DBK starting with a priori codes with subsequent codes emerging from data itself. Emerging codes from each of the two coders were compared, discussed and verified to ensure consistency in the interpretation of emerging codes. Data coding involved moving back and forward between the data set while comparing codes with domains within the Health,

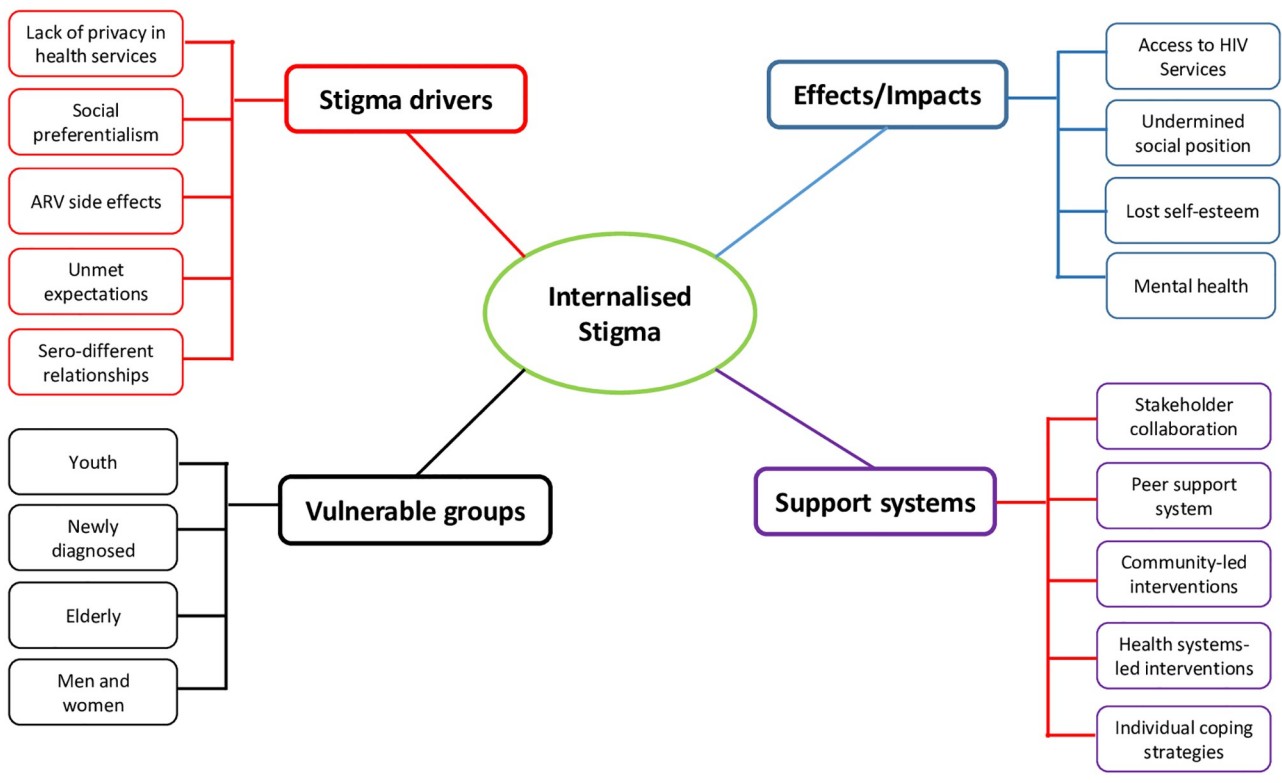

**Fig 2. Emerging themes and sub-categories.**

Stigma and Discrimination Framework while at the same time triangulating analyzed data between the three data sources. Through peer debriefings and discussions during data coding process, it was evident that saturation of information was attained, and this observation was first reached during the data collection process. Searching for themes and sub-themes involved sorting the different coded categories and then integrating them to generate sub-themes and themes (See Fig 2). Afterwards, the research team collectively reviewed and refined candidate themes and sub-themes by reading extracts for each theme and sub-theme to check if they provided a coherent story. After data analysis, results were provided to selected study participants and other stakeholders at the national stakeholder meeting and district levels to provide feedback on the findings. A 32-item checklist of the Consolidated criteria for reporting qualitative research (COREQ) guided the reporting of this study [37].

## Ethics considerations

The study ethical approval was obtained from the Malawi National Commission of Science and Technology (NCST) through its Committee on Social Science and Humanities (Reference No. P.02/21/554). Written or thumbprint informed consent was obtained from all study participants. No participant under the age of 18 years was included in this study.

## Findings

This qualitative inquiry explored the manifestations, drivers and impacts of HIV internalized stigma among people living with HIV in Malawi. A total of 51 individuals participated in this

**Table 2. Study participant's gender and age.**

| Sex | Key informants | Group discussions (n = 3) | Age Range |
|---|---|---|---|
| Male | 14 | 4 | 19–60 |
| Female | 8 | 22 | 19–58 |
| Transgender | | 3 | 26–35 |
| Total | 22 | 29 | |

study as key informants (n = 22) and as FGD participants (n = 29). Demographic characteristics of the study participants is presented in a Table 2.

Qualitative results have been broadly categorized into three themes namely, 1) perceived sub-groups vulnerable to internalized HIV stigma; 2) experiences of internal (both feelings and manifestations) stigma among people living with HIV, and 3) effects of internalized stigma among people living with HIV.

## Perceived vulnerable population sub-groups

Heterogeneous perspectives were expressed by study participants in respect to groups of people deemed to be most vulnerable to internalized HIV stigma. While some study participants submitted that the adolescents or the youth and individuals newly diagnosed with HIV or initiated on ART were highly susceptible to internalized stigma, some maintained that the elderly men and women were most vulnerable to this form of HIV stigma as shown in a quote below:

> *"Women who are probably in discordant relationships can go through that. . ., the ones who are always getting sick. Especially those that have not been diagnosed, those ones I know that sometimes they are even called names by their communities. And when people know they are taking ARVs it's the same.*

(Female, Policy maker, Lilongwe)

For the young boys and girls, their vulnerability to internalized stigma was perceived to be amplified due to vertical transmission of HIV from their parents. Internalized stigma in this group also surfaced because of their wider social networks especially when some of their peers exhibited negative feelings or dislike toward people living with HIV. Furthermore, the youths' high exposure and access to information that shows how others have previously been stigmatised due to their HIV status also resulted into internalizing their feelings about HIV.

> *". . .I can say mainly self-stigma is in youths. . . .A certain number of youths nowadays are HIV positive because of their parents. I am not saying all of them but some. So, that is a challenge because they feel like I could have been negative [HIV negative], but now I am positive [HIV positive] because of my parents. So, the hatred comes there, and another group of the youths are shy because [they feel that] 'I cannot be like this. I am small, I am younger, I cannot be HIV positive'. So, you see that element kills them."*

(FGD, Chisomo CBO, T/A Malemia, Zomba)

In this specific sub-group, more girls than boys were perceived by study participants to shoulder the largest burden of internalized HIV stigma because of their tendency of internalizing experiences and emotions which in turn encouraged self-isolation and social withdraw behaviours as illustrated in the quote below.

*". . . when the children realize that their friend is getting medication at the ART clinic, they start isolating that friend. So, in that way that person gets on with life but in an isolated environment. They don't associate with the other people . . .".*

(Male, Health Provider, Nathenje, Lilongwe District).

In the quote above, external forms of HIV discrimination are portrayed to ignite internalized forms of stigma with this intersection mediated through stigma drivers such as fear of social ramifications, lack of awareness, blame and stereotypes. The quote below demonstrates how externalised HIV stigma taking the shape of isolation of a teenager living with HIV precipitated internalized HIV stigma and depression.

*"In school this happens a lot, we once had a child that was born with HIV, she passed on now. She was treated badly by her friends and the teachers as well. Her friends would not play with her . . ., and the teachers would tell her to sit separately from her friends . . .. She was not leaving a happy life because she was lonely and depressed which contributed to her death at 15 years old 3 years ago".*

(Respondent 1, Female, FSW, FGD, Mwanza)

Some study participants narrated that elderly men and women were highly vulnerable to internalized HIV stigma because their social position and esteem as elders, the community expectation that the elderly are responsible individuals, exemplary and are usually not sexually active (see below).

*"There are older people who are uncomfortable to go to the facility and get their medication because people are going to judge them".*

(Female, KII, Policy maker, MoH).

Our data did not further investigate this dimension from a gender perspective to understand whether or not, a perception of the elderly being vulnerable to internalized stigma was gendered. To other study participants, internalized stigma was said to be mostly experienced by men because of their gendered roles and responsibilities. Men's position as household providers, and the expectation that they should work hard to provide for their families was perceived to encourage them to avoid anything that may undermine their position and portray them as being vulnerable and weak. To avoid being labelled as weak, men were said to internalize negative experiences linked to their HIV status. Similarly, women's lack of power within a household and their dependence on male partners for their livelihood were said to rob them of their assertiveness to discuss HIV-related matters including disclosing their HIV status to a male partner. However, many participants agreed that women were open to discussing sensitive experiences linked to their HIV status to other people including peers and healthcare workers than men, thus finding some form of support to help them navigate their problems.

*". . .most of the times you can see that men are the ones who are really empowered or given some positions in running the affairs of the community. So, in that situation, I think this sometimes encourages or discourages them to access some services related to HIV and AIDS. A lot of men will decide not to go access such services from a team which is composed of women in the majority."*

(Male, Victim Support Unit Officer, Mulanje)

Several participants agreed that HIV mixed-status couples were the most vulnerable to internalized stigma especially the sexual partner who has tested HIV-positive. Anxieties that one partner was not infected with HIV were perceived to be amplified in mixed-status couples as individuals who tested HIV-positive were worried about a potential marriage breakdown or abandonment because of being worried about HIV transmission or sero-sorting tendencies while in these relationships resulting into infidelity (see quote below). Interestingly, few study participants narrated that internalized stigma is not selective and affects people equally.

*"I have never noticed the one who is mostly affected but I guess women. Women who are probably in discordant relationships can go through that maybe women with children, the ones who are always getting sick".*

(Female, KII, Policy maker, DHA, MoH).

## The occurrence, experiences, and available structures of redressing internalized HIV stigma

Study participants agreed that HIV stigma broadly remained common among people living with HIV and emerged from multiple and intersecting fronts such as poverty, gender, age and other social stratifiers. The experience of externalised forms of HIV stigma was said to be more apparent than internalized stigma and manifest in different social groupings within communities including religious groups, social groups, schools, and workplaces. A quote below exemplifies how experienced externalised stigma intersect with or may produce/re-produce internalized forms of HIV stigma as people living with HIV deploy self-isolation as a coping mechanism to externalised stigma.

*". . .it also happens that some people stop socializing with their friends because they feel that they can't have friends since they have HIV, they isolate themselves".*

(FGD, Chisomo CBO, T/A Malemia, Zomba).

Life Story 1 below typifies the extent to which internalized stigma propels an individual living with HIV to feeling helpless and compels him/her to conceal from the society through self-isolation. The life story also illuminates the impotency of available support structures; the family and community leadership in addressing latent forms of stigma.

The young girls and boys who participated in this study indicated that internalized HIV stigma affected their school attendance as they had previously witnessed peers dropping out of school because of depressive tendencies arising from internalized stigma (see quote below).

*"Some students stop going to school because they are afraid of being teased at school about their status. So, some students decide it is not worth going through that and stop going to school".*

(FGD, Chisomo CBO, T/A Malemia, Zomba)

The quote above demonstrates how experiencing HIV stigma and discrimination in schools such as ridicule, bullying, gossips among others against young boys and girls produced or reproduced internalized stigma which in turn triggered other social consequences in this case diminished school attendance and school drop-out. A key informant from Malawi's Ministry of Education acknowledged that the education system fails to address HIV stigma issues

### Life Story 1: Self-isolation as a coping mechanism from external stigma

Patrick is a chairperson of a Community Based Organization, and holds senior positions within two reputable institutions in Thyolo and Blantyre districts. Despite having social positions that place him in front of people, Patrick's main desire is to 'create boundaries of seclusion" for himself. He feels he is suffering a lot because other people are gossiping, castigating and stigmatizing him because of his HIV status even by his own family. He tried several times to bring his concerns to the attention of traditional leader in his village but has not received remedy and in the process has completely lost trust in the available system for addressing HIV stigma in the community. He says despite his knowledge on law for redress; the difficulty is that the problem is within his immediate family. He feels he is very insecure. It's the more reason he wants to "create own wall/barrier" for self-protection so that others cannot reach him. Patrick feels so helpless about his situation.

including self-stigma among pupils living with HIV primarily because educators have limited knowledge, skills, and tools. Furthermore, the Malawian school curriculum was said to be incapable of providing adequate information on dangers of internalized stigma in schools as expressed by the following quote:

> *"Some of the boys are also dropping out of school simply because some of their friends were laughing at them when they were seen receiving treatment [ART] in room 6".*

(REAP, KII, Rumphi)

Community experiences of internalized stigma was said to also happen within a family context. Stigmatization of people living with HIV by family members, especially by individuals who do not accept to be identified with someone who is living with HIV, create internalized stigma. Non-existent systems of redress and lack of psycho-social support create an environment where it is common for individuals to both internalise negative experiences related to ones HIV status and cope with HIV stigma through self-isolation have been illustrated by the following quote:

> *". . . even my relatives were discriminating me. Sometimes they would refuse to eat in the same plate with me saying they might contract the disease. Some would take my properties like land . . .. When I told them this land is from my mother, they answered by saying you are already dead, you cannot have the land. It was so hard for me. . .. The time I was pregnant, they said you are not going to be able to feed that child because HIV and /AIDS is already eating the child inside the womb. . ."*

(Respondent, FGD with people living with HIV, Neno)

These quotes are emblematic reactions from study participants and demonstrate the experience of externalised stigma manifesting through verbal abuse, physical land grabbing and rejection by other family members that trigger self-isolation (see quote below).

### Life Story 2: Male partner abandonment after disclosure

Zione is currently a sex worker in Mwanza district. Before she became aware of her HIV-positive status she was a happy person. But she became very worried when she received her HIV-positive result after testing. "I feel painful to be on drugs (ARVs) and I always worry about the future of my children". Zione's further narrated her ordeal which she referred to as a "a mistake"; the experience of separating from her long, stable and supportive male partner. It all started when she avoided to notify her partner that she was on ART. When her partner accidentally discovered ARV drugs in their bedroom, it created a major rift between them. Although Zione tried hard to ask for forgiveness, her male partner rejected her plea saying, 'how come I discovered the drugs on my own without you sharing this information with me?' From that time, Zione's relationship was severed, and she was devastated with their separation.

*"If I can go to complain, but in my situation and if I report these people, I am talking about they are my relatives, so if I report them, I will be in danger".*

(Male, PWD, IDI, Thyolo).

In sexual relationships, several participants narrated that male partners normally terminate relationships after learning for the first-time that their female partner is living with HIV (See Life Story 2 below). For both men and women who are living with HIV who are not married or in sexual relationship, internalized HIV stigma was often related to the anxiety of finding a partner who would be willing to love them unconditionally.

Participants were afraid that their HIV status might negatively affect future aspirations of having biological children (see Life Story 3). It was apparent that people living with HIV included in this study were not aware of undetectable equals untransmittable (U = U) information. The quote illustrates elements of internalized stigma around prospects for sexual relationships and having biological children.

*"Now there is the youth who want to get married and even when a man wants to marry a new wife though he is HIV+; they don't want the other person to know".*

(Person living with HIV IDI Rumphi

A highly ranked policy maker echoed sentiments expressed by people living with HIV that internalized stigma is usually prevalent among HIV mixed-serostatus couples. In these couples, the partner who is living with HIV usually develop internalized stigma because of the anticipation of rejection or actual abandonment by the partner who is HIV negative (see Life Story 3 above). Actions by the negative partners such as rejection and exclusion often trigger internalized stigma within the partner who is living with HIV (see Quote below).

*But I know of families where you have sero-different couples, sometimes the other partner feels stigmatized by the other. Based on the behaviors in some instances it is true, it is actual stigma itself not by the individual but from the partner".*

(Female, Policy Maker, DHA, MoH)

### Life Story 3: Prospects to having own children

Mabvuto is a person living with HIV and disability from Thyolo district. He has no problem accepting his status as a person with living with HIV and disability. His main problem is what other people in the community gossip about people living with both disability and HIV. He narrates that given his HIV status; he "... cannot marry because no one can be in a long-term sexual relationship with a person who is taking ARVs". He says it's hard for him to express himself in an environment where other people's knowledge of his HIV status brings more negative issues to his life. He is visibly afraid, and this fear makes him uncomfortable to stand on an ART queue at a health facility which is not fenced and yet close to a roadside marketplace. This discomfort makes him shy and brings him sense of unworthiness even as he feels it is better for him to stop taking ARVs and die.

## Internalized HIV stigma, self-isolation and mental health

Most study participants agreed that the experience of internalized HIV stigma affected people living with HIV self-esteem and quality of life in different ways. At the individual level, internalized forms of HIV stigma were seen to affect people living with HIV's desire and ability to establish social relationships including finding a sexual partner and enjoying a 'normal life'. Life story 2 depicts how internalized HIV stigma diminished an individual's ability to establish social relationships with other people such as colleagues and sexual partners. The use of self-isolation as an individual coping mechanism and the internalization of negative experiences linked to their HIV status seemed to disempower individuals to seek life-saving healthcare. People living with HIV usually felt helpless when experiencing life challenges including externalised forms of stigma and had little trust in the available systems of redress or of other care for internalized stigma such as peer-support, community-level and formal health interventions. Some people living with HIV felt embarrassed or uncomfortable with their HIV status even without anyone projecting externalised forms of stigma on them. A representative quote below demonstrates the social and economic negative impacts of internalized HIV stigma on access to care and treatment within established facilities.

> *"There are some who are embarrassed. It happens that people choose to go to facilities that are very far away so that the people who know them are not aware of their condition".*

(Respondent 3, FGD, Chisomo CBO, T/A Malemia, Zomba).

Internalization of negative experiences linked to HIV-positive status seemed to shape how people living with HIV interacted with other people and dealt with life challenges as shown in life stories 1 and 2 above. People living with HIV seemed to live a precarious life, as they were constantly unclear about their future (See Life Story 3 above). The fact that some people living with HIV considered themselves as unworthy (see a quote below); they were not afraid to take risks such as living in isolation, dropping out of school and avoiding life-saving ART.

> *"When I tested HIV positive, I felt as though I am not worthy of anything,"*

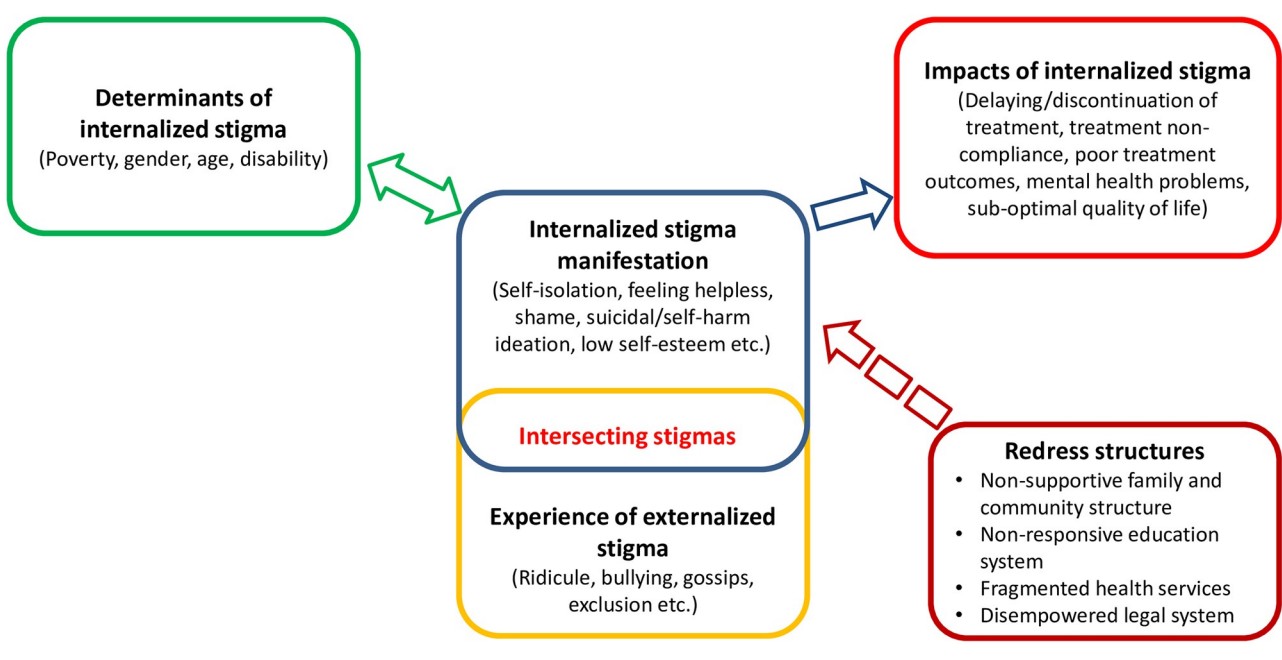

**Fig 3. Mapping internalized stigma in Malawi.**

(Person living with HIV IDI Rumphi)

Thus, internalized HIV stigma was described by people living with HIV as a potential source of mental health problem such as anxieties and depression. The quote below shows the relationship between externalized stigma and internalized stigma and how this work together to create mental health problems.

*". . . most of the people who are stigmatized fail to take their medication as required. Sometimes when people are stigmatised, they feel very anxious, they are sad, and they don't have the appetite to take the medication. They may even fail to go to the next visit at the ART clinic to take their medication because they are stigmatized".*

(Male, Health Provider, Nathenje, Lilongwe District).

At relationship level, people living with HIV were said to experience problems in staying in new sexual relationships after disclosing their HIV status to the partner as demonstrated by Life-story 3 above.

Fear of consequences linked to disclosure of HIV status promoted non-disclosure tendencies which is not ideal for HIV prevention efforts. Internalized stigma was also said to affect access to and utilization of life-saving treatment for HIV and contributed to poor treatment adherence among people living with HIV. Study participants agreed that individuals with HIV-related internalized stigma were generally afraid of being seen queuing for HIV-related services by other community members when accessing HIV treatment and care at a health facility (see quote below).

*". . . some people fail to access ARVs because of issues of stigma and discrimination, they fear that when they are a lot of people at the health centre, they might be seen going to an ART clinic".*

(Social Services Officer, KII, Ntcheu)

## Discussion

This study investigated the manifestations, drivers and impacts of HIV internalized stigma among people living with HIV in Malawi. The study shows divergent perspectives on groups of people perceived to be most vulnerable to internalized HIV and that internalized stigma intersect with externalised stigma as well as other social stratifiers (e.g., age, gender, disability etc.). This intersection is mediated by stigma drivers such as fear of social consequences, lack of awareness, blame and stereotypes. Furthermore, the study illustrates that internalized stigma impacts quality of life and may influence access to health care services (See Fig 3).

Research has historically tended to focus on external forms of HIV stigmas and has informed HIV stigma reduction programming which, in most part, with little consideration given to how various stigma types might impact strategy choices and outcomes [38]. The findings of this study are crucial for reigniting a meaningful discussion of HIV stigma in its wider scope, in order to generate robust interventions that broadly addresses this global problem. Framed within the Health, Stigma and Discrimination Framework [33], this study has illustrated that internalized HIV stigma is driven largely by the fear of social ramifications, social judgement, social stereotypes and prejudices, gender norms, cultural expectations and the unavailability of mechanisms of support and redress.

Internalized HIV stigma as shown in this study seem to be experienced mostly through perceived and anticipation of external forms of stigma and its manifestation is usually characterized by self-stigmatizing behaviors through social isolation or withdrawal. Some examples of the outcomes from the experience of internalized HIV stigma among people living with HIV include lack of access to social justice, compromised enjoyment of health rights, and poor adherence to life-saving treatment. Despite Malawi's efforts to address HIV stigma broadly, we have shown through our findings that the country lacks preventive and mitigation measures that are specific to internalized HIV stigma and this perpetuates and intensifies effects of this form of stigma. In Nigeria, a study reported a high proportion of internalized stigma among people living with HIV (31.4% among people in peer groups and 30.2% among people without peer groups) and inadequate availability of mitigation intervention to manage it [39]. Helms (2017) further recognizes that individuals with internalized HIV stigma usually have limited interpersonal resources at their disposal to manage their condition [40].

Findings from this study have demonstrated that exposure to externalized forms of stigma further fuels internalized stigma among people living with HIV and the acceptance of stigmatizing stereotypes linked to HIV (see Fig 2). Fazeli et al (2017) also reported the intersection between HIV-related discrimination and the internalization of HIV stigma [41]. The acceptance of what some scholars have termed a 'spoiled identity' legitimizes and normalizes internalized HIV stigma which, in this study, were discernible through self-protective tendencies such as self-exclusion, social retraction, avoidance of actions that may unveil ones HIV-positive and risk-taking behaviours such as poor access to HIV health services and suboptimal compliance to ART which have been reported elsewhere [42,43]. A study in Uganda found that the worries about being at an ART clinic and internalized loss of hope due to fear of death were important impediment to same-day ART initiation by female sex workers [44]. In Malawi, the experience or anticipation of marriage breakup or abandonment within HIV mixed-status couples was reported to fuel anxieties about future prospect of having an established sexual partner or biological children [45]. Clearly, the predominant social identities, labels, and stereotypes connected to HIV contributed to the internalization of stigmatizing attitudes which in turn undermines adaptive coping and social connection.

In a context of deficient enforcement of existing laws and policies that protect people living with HIV from internalized HIV stigma, the privation of knowledge on availability and effectiveness of the support and redress system left people living with HIV to develop alternative and unconventional strategies of coping with internalized HIV stigma. These strategies include self-isolation, delaying or discontinuation of treatment, non-compliance to treatment, and the use of multiple health facilities and two health record books with one being used exclusively on HIV health care and the other one for the other medical conditions apart from HIV. Use of some of these coping strategies have also been reported in both low and high income settings [46–49]. Our findings confirm that the way health facilities are presently organized may aggravate the internalization of HIV stigma among people living with HIV since there is little or nothing to offer to address internalized HIV stigma.

Although externalized forms of HIV stigma have been categorized as different from internalized stigma, this study has shown that these forms of stigmas are somewhat interconnected, with one form triggering the other. The exposure to or experience of externalized forms of HIV stigma have also been noted to gradually get internalized and normalized by people living with HIV through perceptions, anticipation and experience [50]. Self-isolation following experience of and/or fear of externalized HIV stigma suggest an interplay of different forms of HIV stigma but also a reproduction and/or transformation of one form of stigma into another [51]. Despite the blended reaction on perceived vulnerable groups to internalized HIV stigma, our findings suggested a potential interaction between internalized HIV stigma with other axes of vulnerability and marginalization such as age, gender, and socio-economic position. Shen et al (2019), reported experiences of internalized HIV stigma intersecting with age and sexual orientation/identities [52].

Finally, our study findings portray the intersection between internalized stigma and health outcomes such as mental health, uptake of services and personal well-being [15,53]. For mental health, internalized stigma have been linked to common mental disorders such as anxieties, depression, suicidal ideation and attempted self-harm and antisocial personality disorders [54]. The layering of internalized HIV stigma with externalised stigma and other axes of vulnerability adds complexities to the analysis of experiences and management of this form of HIV stigma. The key methodological limitation of this study is the unavailability of voices from other important groups of people such as male sex workers. In addition, the study has mostly dealt with internalized HIV stigma as it is experienced within the Malawian context which may be different to other countries within the sub-Sahara African region.

## Conclusion

This paper analyzed experiences of internalized HIV stigma among people living with HIV in Malawi guided by the Health Stigma and Discrimination framework in order to understand its occurrence and implications. Characterization of internalized HIV stigma is key to formulation of targeted and context specific solutions to broadly mitigate this problem in all its forms. This study has demonstrated that internalized HIV stigma is pervasive in Malawi and intersects with other forms of stigma to affect the quality of people living with HIV life and their health outcomes. Study findings suggest that the failure to address internalized forms of HIV stigma may potentially constitute a human rights violation, jeopardizing physical and mental well-being of people living with HIV and people who depend on them. Therefore, addressing internalized stigma is an important element of dealing with both externalised forms of HIV stigma and discrimination and improving health outcomes among people living with HIV.

## Supporting information

**S1 Appendix. Data collection tools.**
(DOCX)

**S1 File.**
(DOCX)

## Acknowledgments

The authors thank study participants; gate keepers and community leaders for permission to carry out the study; the study team members including the data collection team namely Khazgani Nasasara, Hastings Mwanza, Kondwani Chirwa, Thandi Msukuma; The Malawi Ministry of Health; The Malawi Liverpool Welcome Trust Clinical Research Programme; The Helse Nord TB Initiative of the Kamuzu University of Health Sciences; Network of Journalists Living with HIV' Prime Health Consulting; Watipa, Australia and The International Aids Society.

## Author Contributions

**Conceptualization:** Moses Kelly Kumwenda, David Black Kamkwamba, Maureen Leah Chirwa, Kasoka Kasoka, Tessa Oraro-Lawrence, Lucy Stackpool-Moore.

**Data curation:** Moses Kelly Kumwenda, David Black Kamkwamba.

**Formal analysis:** Moses Kelly Kumwenda, David Black Kamkwamba, Maureen Leah Chirwa, Kasoka Kasoka, Tessa Oraro-Lawrence, Lucy Stackpool-Moore.

**Funding acquisition:** Kasoka Kasoka, Tessa Oraro-Lawrence, Lucy Stackpool-Moore.

**Investigation:** Moses Kelly Kumwenda, David Black Kamkwamba, Maureen Leah Chirwa.

**Methodology:** Moses Kelly Kumwenda, David Black Kamkwamba, Maureen Leah Chirwa, Kasoka Kasoka, Tessa Oraro-Lawrence.

**Project administration:** Moses Kelly Kumwenda, David Black Kamkwamba.

**Resources:** Moses Kelly Kumwenda, David Black Kamkwamba, Maureen Leah Chirwa, Kasoka Kasoka, Tessa Oraro-Lawrence, Lucy Stackpool-Moore.

**Software:** Miriam Taegtmeyer.

**Supervision:** Moses Kelly Kumwenda, David Black Kamkwamba, Maureen Leah Chirwa, Kasoka Kasoka, Miriam Taegtmeyer, Tessa Oraro-Lawrence, Lucy Stackpool-Moore.

**Validation:** Miriam Taegtmeyer.

**Writing – original draft:** Moses Kelly Kumwenda, David Black Kamkwamba, Maureen Leah Chirwa, Kasoka Kasoka, Miriam Taegtmeyer, Tessa Oraro-Lawrence, Lucy Stackpool-Moore.

**Writing – review & editing:** Moses Kelly Kumwenda, David Black Kamkwamba, Maureen Leah Chirwa, Kasoka Kasoka, Miriam Taegtmeyer, Tessa Oraro-Lawrence, Lucy Stackpool-Moore.

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
