## [Decision Letter · Decision Letter 0]

29 Nov 2022

PONE-D-22-27597“A peer-supporter gives us advice on how we can live happily with HIV” - The manifestations and coping mechanisms of internalised HIV stigma among people living with HIV in MalawiPLOS ONE

Dear Dr. Kumwenda,

Thank you for submitting your manuscript to PLOS ONE. After careful consideration, we feel that it has merit but does not fully meet PLOS ONE’s publication criteria as it currently stands. Therefore, we invite you to submit a revised version of the manuscript that addresses the points raised during the review process.

We look forward to receiving your revised manuscript.

Kind regards,

Nelsensius Klau Fauk, S.Fil., M., MHID, MSc, PhD

Academic Editor

PLOS ONE

Journal Requirements:

3. You indicated that you had ethical approval for your study. Please clarify whether minors (participants under the age of 18 years) were included in this study. If yes, in your Methods section, please ensure you have also stated whether you obtained consent from parents or guardians of the minors included in the study or whether the research ethics committee or IRB specifically waived the need for their consent.

6. Please amend either the abstract on the online submission form (via Edit Submission) or the abstract in the manuscript so that they are identical.

Additional Editor Comments:

**Introduction**

Lines 1-2: “HIV stigma remains pervasive at each point of the HIV care continuum and is profoundly ingrained within social, economic, cultural and historical contexts”

Use and cite the following current resources to support this statement:

Psychological and social impact of HIV on women living with HIV and their families in low-and middle-income Asian countries: A systematic search and critical review. International Journal of Environmental Research and Public Health 19 (11), 6668. https://www.mdpi.com/1660-4601/19/11/6668

Stigma and discrimination towards people living with HIV in the context of families, communities, and healthcare settings: a qualitative study in Indonesia. International journal of environmental research and public health 18 (10), 5424. https://www.mdpi.com/1660-4601/18/10/5424

Second paragraph: “Globally, the 2020 UNAIDS Global AIDS Update observed that HIV stigma persist to obstruct people living with HIV from making use of HIV related services and that discriminatory attitudes towards people living with HIV continue to be prevalent in many countries with the available legal systems unable to provide protection (11).”

Use and cite the following current resources to support this statement:

HIV stigma and moral judgement: qualitative exploration of the experiences of HIV stigma and discrimination among married men living with HIV in Yogyakarta. International Journal of Environmental Research and Public Health 17 (2), 636. https://www.mdpi.com/1660-4601/17/2/636

HIV stigma and discrimination: perspectives and personal experiences of healthcare providers in Yogyakarta and Belu, Indonesia. Frontiers in medicine 8, 625787. https://www.frontiersin.org/articles/10.3389/fmed.2021.625787/full

Fifth paragraph: “The internalisation of stigma have been reported to affect quality of life and health outcomes including mental health issues such as anxiety and depression (20), access to HIV-related services (21), adherence to antiretroviral treatment (ART) (22) and willingness to disclose (23).”

Use and cite the following current resource to support this statement:

Risk Factors and the Impact of HIV among Women Living with HIV and Their Families in Yogyakarta and Belu District, Indonesia. PhD Disertation, Flinders University, Adelaide, Australia. (Look at section 5.3.1; 5.4; 6.3.1; 6.4 and chap 7). https://flex.flinders.edu.au/file/eac1a04e-a24b-4888-ace8-50f06805069e/1/Fauk%20Thesis2022.pdf

Sixth Paragraph: “Despite the growing debate on getting to the heart of HIV stigma (7,25), researchers have traditionally focused on overt forms of stigma (26,27) without paying attention to internalised forms of stigma that are subtle and operates under the surface. We explored the manifestations, drivers and impacts of HIV internalised stigma among people living with HIV in Malawi.”

Please look at the section of the dissertation above and revised this gap because there have been many studies exploring internalised stigma.

**Analytical approach**

The authors described the data analysis process they have undertaken, but the steps are not clear to follow. I strongly suggest authors to consult data analysis section of the following resources and cite them to support their description data analysis:

Structural, Personal and Socioenvironmental Determinants of HIV Transmission among Transgender Women in Indonesia. *Int. J. Environ. Res. Public Health* **2021**, *18*, 5814. https://doi.org/10.3390/ijerph18115814
Risk Factors and the Impact of HIV among Women Living with HIV and Their Families in Yogyakarta and Belu District, Indonesia. (Data analysis section: Page 109-114). https://flex.flinders.edu.au/file/eac1a04e-a24b-4888-ace8-50f06805069e/1/Fauk%20Thesis2022.pdf

**Reference**

Please revise your reference, look at the journal guidelines about referencing style. There are many mistakes in the references.

Reviewers' comments:

Reviewer's Responses to Questions

**Comments to the Author**

1. Is the manuscript technically sound, and do the data support the conclusions?

Reviewer #1: Partly

Reviewer #2: Yes

2. Has the statistical analysis been performed appropriately and rigorously? 

Reviewer #1: N/A

Reviewer #2: N/A

3. Have the authors made all data underlying the findings in their manuscript fully available?

Reviewer #1: No

Reviewer #2: Yes

4. Is the manuscript presented in an intelligible fashion and written in standard English?

Reviewer #1: Yes

Reviewer #2: Yes

5. Review Comments to the Author

Reviewer #1: Comments to the Authors

Dear authors,

Thank you for the chance to offer feedback. I thoroughly enjoyed your article and believe that paper is worth pursuing.

I have offered some feedback below. I hope you find my comments/suggestions helpful.

INTRODUCTION

- “More than a third of people living with HIV (PLHIV) report having been physically assaulted…” was written/reported in 2012.

As this is a prevalence report, better to provide more recent data/references.

- “A recent systematic review and meta-analysis reported significant associations between reporting stigma and having CD4 counts of less than 200….” Great to read this sentence as solid evidence to state the seriousness of the problem. However, the authors need to note whether the stigma mentioned here includes the internal stigma.

- In the first paragraph, the authors mentioned stigma in Malawi then moved to global issues in the second paragraph. Again, in the third paragraph, the authors discussed the HIV stigma in Malawi. Following this paragraph, the authors explained about stigma. The authors need to revise the outline of the introduction part to make a better flow.

METHODS

- State the reporting guideline used by the authors.

- Most qualitative reporting guidelines (i.e., COREQ) require an explanation related to the credentials of the researchers who collected the data (FGDs and/or IDIs). I could read of it in the part of IDI, but no information for the FGDs. Also, who did the analysis (not only the initials); gender, and credentials? Please explain more, also the statement of “….experienced qualitative researcher” and “…by a team of skilled qualitative research assistants” (i.e., qualitative course, research, years?).

- I prefer the use of “Participants” section instead of “Sample size” due to the quantitative sound of the sub-title (also in table 1).

- “Data collection tools (KII and FGD guides) were translated into local languages (Chichewa and Chitumbuka) and back into English; piloted to check their reliability and validity.”

How did the authors check the reliability and validity of the data collection tools (KII and FGD guides)?

- How did the authors develop the IDI and FGD guidelines?

- Who did the translation into English of the transcripts? How did the authors ensure the validity of the translation?

- Aside from methodology and participant triangulation, are there any other triangulation approaches used by the authors?

- The third sentence of “Participant recruitment and data collection” is too long. Separate into more sentences or use (1), …. (2)…. etc., for a better read.

- Analytical approach: what qualitative data analysis was used by the authors? (thematic?)

- How did the team decide on the data saturation? (i.e., peer debriefing?).

- Time of data collection?

FINDINGS

- It would be helpful if the authors provided a table or a narrative way to describe the characteristics of the participants (i.e., gender, age, etc.).

- The authors may provide the codes, categories, sub-themes, and themes developed from the study. A coding tree or table will be beneficial to understand the process.

- The objective of this study was “We explore the manifestations, drivers, and impacts of HIV internalized stigma among people living with HIV in Malawi. Understanding experiences of internalized stigma is the first step to determine befitting mitigation options broadly”.

I can relate the first theme of “perceived sub-groups vulnerable to internalized HIV stigma” to answer the study's purposes. But the second and third themes need to restructure as the article only repeated the study objectives.

I.e. (1) objective: “Understanding experiences of…”. Theme: “Experiences of

internal (both feelings and manifestations) stigma among people living with HIV”.

(2) Objective: “impacts of HIV internalized stigma…”. Theme: “Effects of

internalised stigma…”. A coding tree will help the readers understand how these themes developed.

DISCUSSION

- “A peer-supporter gives us advice on how we can live happily with HIV”. The

manifestations and coping mechanisms of internalised HIV stigma among people living with HIV in Malawi.

After reading the whole paper, I did not understand why the authors stated this quote in the title. If this quote is highlighted in the title, it should be the “heart” of the findings and discussion parts. Moreover, in the title authors mentioned “The manifestations and coping mechanisms”. Did this quote represent the manifestations and coping mechanisms?

- Strength and limitations? i.e., as the authors did not conduct the member checking.

Reviewer #2: First, I would like to thank and congratulate those who are writing this document, since carrying out a study of the problems that are experienced at an institutional and national level seems to me to be an action of resistance, and at the same time of intelligence , by using platforms such as scientific journals (of privilege) to make these latent problems visible today.

It is for the above that the article is built with highly relevant and valuable content to be published for worldwide. Since the speeches clearly capture traces of realities that are oppressed, excluded, and silenced.

Second, details are missing on aspects of the design. How were the participants selected? Why were the areas of the country chosen? Why sex workers? How old are the participants? At no time is the age range mentioned. These are critical methodological details required... can you provide details? Likewise, as a fundamental point, they do not mention from which paradigm or epistemology, approach or perspective they are located to analyze the investigative study.

Finally, can your discussion provide a more detailed discussion on what is the main contribution of your study to the existing literature and what are the implications of your study and how it differs from other studies already done?

Lastly, what psychosocial intervention proposal can be built to deliver solutions to the problems that have emerged?

6. PLOS authors have the option to publish the peer review history of their article (what does this mean?). If published, this will include your full peer review and any attached files.

Reviewer #1: No

Reviewer #2: **Yes: **Estefany Castillo Ravanal

---

## [Author Response · Author response to Decision Letter 0]

20 Feb 2023

Reviewers' comments:

Reviewer's Responses to Questions

Comments to the Author

1. Is the manuscript technically sound, and do the data support the conclusions?

Reviewer #1: Partly

Reviewer #2: Yes

Response: We have strengthened our conclusion by drawing much from the data. 

2. Has the statistical analysis been performed appropriately and rigorously?

Reviewer #1: N/A

Reviewer #2: N/A

3. Have the authors made all data underlying the findings in their manuscript fully available?

Reviewer #1: No

Reviewer #2: Yes

Response: The minimal data set underlying the results described in this manuscript can now be found at the Qualitative Data Repository (https://doi.org/10.5064/F6UJUEP3). 

4. Is the manuscript presented in an intelligible fashion and written in standard English?

Reviewer #1: Yes

Reviewer #2: Yes

5. Review Comments to the Author

Reviewer #1: Comments to the Authors

Dear authors,

Thank you for the chance to offer feedback. I thoroughly enjoyed your article and believe that paper is worth pursuing.

I have offered some feedback below. I hope you find my comments/suggestions helpful.

INTRODUCTION

- “More than a third of people living with HIV (PLHIV) report having been physically assaulted…” was written/reported in 2012.

As this is a prevalence report, better to provide more recent data/references.

Response: We included more recent references and have revised the sentence accordingly. 

- “A recent systematic review and meta-analysis reported significant associations between reporting stigma and having CD4 counts of less than 200….” Great to read this sentence as solid evidence to state the seriousness of the problem. However, the authors need to note whether the stigma mentioned here includes the internal stigma.

Response: Yes, the stigma mentioned in the sentence include internal stigma. Authors of the paper investigated many dimensions of stigma including internalized stigma described in the paper as self-image, shame, internalized stigma etc. We have revised the sentence to demonstrate that internalized stigma is included. 

- In the first paragraph, the authors mentioned stigma in Malawi then moved to global issues in the second paragraph. Again, in the third paragraph, the authors discussed the HIV stigma in Malawi. Following this paragraph, the authors explained about stigma. The authors need to revise the outline of the introduction part to make a better flow.

Response: We have revised the outline of the introduction section to improve the flow of the paper.

METHODS

- State the reporting guideline used by the authors.

Response: We have mentioned COREQ as a framework that guided our reporting.

 Most qualitative reporting guidelines (i.e., COREQ) require an explanation related to the credentials of the researchers who collected the data (FGDs and/or IDIs). I could read of it in the part of IDI, but no information for the FGDs. Also, who did the analysis (not only the initials); gender, and credentials? Please explain more, also the statement of “….experienced qualitative researcher” and “…by a team of skilled qualitative research assistants” (i.e., qualitative course, research, years?).

Response: We have included the credentials of individuals who were involved in data collection and analysis in the methods section.

- I prefer the use of “Participants” section instead of “Sample size” due to the quantitative sound of the sub-title (also in table 1).

Response: We have replaced the word ‘sample size’ with ‘participants’ as suggested by the reviewer

- “Data collection tools (KII and FGD guides) were translated into local languages (Chichewa and Chitumbuka) and back into English; piloted to check their reliability and validity.” How did the authors check the reliability and validity of the data collection tools (KII and FGD guides)?

Response: The sentence has been revised to clarify the aim of pilot testing data collection tools. The sentence now reads ‘Data collection tools (KII and FGD guides) were translated into local languages (Chichewa and Chitumbuka) and back into English; piloted to check their reliability in capturing data that addressed the research objectives.’ We checked if the tools were capturing the information that we wanted to get in order to answer the research questions.

- How did the authors develop the IDI and FGD guidelines?

Response: In the methods section, we have included the following sentence that describes how the data collection guides were developed. “Data collection tools (KII and FGD guides) were developed by MK, DK and MC. The development of the guides was framed by the study research questions which were aligned to the International AIDs Society’s’ (IAS) agenda on “Getting to the Heart of Stigma” and shaped by the Health, Stigma and Discrimination Framework.”

- Who did the translation into English of the transcripts? How did the authors ensure the validity of the translation?

Response: Members of the research teams included a mixture of individuals fluent in Chichewa and Chitumbuka languages. Please note that Chichewa is one of the national languages in Malawi and as such, most Malawians understands this language. In terms of translation, Hastings Mwanza translated from English to Tumbuka. Khazgani Nasasara translated from Chichewa to English. David Kamkwamba proof read the Chichewa translation while Maureen Chirwa proof read the Tumbuka translation. We hoped that these steps were sufficient to ensure the validity of our data collection tools. 

- Aside from methodology and participant triangulation, are there any other triangulation approaches used by the authors?

Response: There were no other triangulation done apart from those described in the methods section. However, a comparison of the new codes was done between two coders involved in the coding process. 

- The third sentence of “Participant recruitment and data collection” is too long. Separate into more sentences or use (1), …. (2)…. etc., for a better read.

Response: The sentence has been separated into two sentences. It now reads: “Participants for the initial sample were purposively selected based on their virtue of being people living with HIV, living with disability and HIV and; being national/, district, and/ community stakeholders with roles and responsibilities in the development and implementation of the legal and policy instruments and, guidelines. Additional participant’s eligibility criteria included; being involved in the allocation of resources for assisting people living with HIV; healthcare providers involved in interventions designed to mitigate effects of HIV stigma; and those involved in fostering legal redress against HIV-related human rights violations.”

- Analytical approach: what qualitative data analysis was used by the authors? (thematic?)

Response: We have clarified this in the analytical approach section. Thematic approach was used to analyse data. 

- How did the team decide on the data saturation? (i.e., peer debriefing?).

Response: We have clarified this in the analytical approach section that ‘data saturation’ was decided at two levels 1) during data collection and 2) during data analysis. At both levels, peer debriefing was used to make this determination. 

- Time of data collection?

Response: We have included the data collection period in the methods section of the manuscript.(6th to 12th April 2021)

FINDINGS

- It would be helpful if the authors provided a table or a narrative way to describe the characteristics of the participants (i.e., gender, age, etc.).

Response: We have provided the characteristics of the participants in Table 2 in the introductory part of the finding section.

- The authors may provide the codes, categories, sub-themes, and themes developed from the study. A coding tree or table will be beneficial to understand the process.

Response: A coding tree from NVIVO nodes has been provided as Figure 3. The figure illustrate the codes that were generated and merged to generate themes

- The objective of this study was “We explore the manifestations, drivers, and impacts of HIV internalized stigma among people living with HIV in Malawi. Understanding experiences of internalized stigma is the first step to determine befitting mitigation options broadly”.

I can relate the first theme of “perceived sub-groups vulnerable to internalized HIV stigma” to answer the study's purposes. But the second and third themes need to restructure as the article only repeated the study objectives.

I.e. (1) objective: “Understanding experiences of…”. Theme: “Experiences of

internal (both feelings and manifestations) stigma among people living with HIV”.

(2) Objective: “impacts of HIV internalized stigma…”. Theme: “Effects of

internalised stigma…”. A coding tree will help the readers understand how these themes developed.

Response: We have provided a coding tree generated from NVIVO as Figure 3. We have also included the main aim of the study as described in the study protocol and clarified what this paper attempts to achieve. We have also revised the themes subtitles of the findings section.

DISCUSSION

- “A peer-supporter gives us advice on how we can live happily with HIV”. The

manifestations and coping mechanisms of internalised HIV stigma among people living with HIV in Malawi.

After reading the whole paper, I did not understand why the authors stated this quote in the title. If this quote is highlighted in the title, it should be the “heart” of the findings and discussion parts. Moreover, in the title authors mentioned “The manifestations and coping mechanisms”. Did this quote represent the manifestations and coping mechanisms?

Response: We have revised the title of the manuscript. It now reads “Lived experiences of People Living with HIV - A qualitative exploration of the manifestation, drivers and effects of Internalised HIV Stigma within the Malawian context. We hope that the new title is broad to cover the full scope of the manuscript.

- Strength and limitations? i.e., as the authors did not conduct the member checking.

Response: We have provided a statement on methodological limitation of the study in the last paragraph of the discussion section.

Reviewer #2: First, I would like to thank and congratulate those who are writing this document, since carrying out a study of the problems that are experienced at an institutional and national level seems to me to be an action of resistance, and at the same time of intelligence , by using platforms such as scientific journals (of privilege) to make these latent problems visible today.

It is for the above that the article is built with highly relevant and valuable content to be published for worldwide. Since the speeches clearly capture traces of realities that are oppressed, excluded, and silenced.

Response: We thank our reviewers for the wonderful comment on our manuscript.

Second, details are missing on aspects of the design. How were the participants selected? Why were the areas of the country chosen? Why sex workers? How old are the participants? At no time is the age range mentioned. These are critical methodological details required... can you provide details? Likewise, as a fundamental point, they do not mention from which paradigm or epistemology, approach or perspective they are located to analyze the investigative study.

Response: Participants were purposively sampled. Participants were selected by virtue of their being people living with HIV and disability, national, district and community stakeholders with roles and responsibilities in development and implementation of the legal and policy instruments, guidelines; those engaged in allocation of resources for assisting people living with HIV; health care providers involved in interventions which are designed to mitigate the effects of specific internal stigma mechanisms tied to specific health behaviors and outcomes and those involved in facilitating legal redress against human rights violations amongst people living with HIV and disability. Key informant interviews, Focus Group Discussions and life stories were used to gain insights on how people living with HIV and disability stigmatize themselves and how they interpret the efforts on stigma reduction in the existing laws, policies, guidelines and structures/systems. Existing collaboration and networks within the team was used to engage networks of people living with HIV across affected population groups, geographical divide of the disease burden, cultural and ethnic diversity in Malawi to draw the participants.

Finally, can your discussion provide a more detailed discussion on what is the main contribution of your study to the existing literature and what are the implications of your study and how it differs from other studies already done?

Response: We have included a discussion of the main contribution and implications of study findings in the second paragraph of the discussion section.

Lastly, what psychosocial intervention proposal can be built to deliver solutions to the problems that have emerged?

Response: No psychosocial intervention directly emerged from this study in particular. But as researchers, the findings from this study have helped us to start conducting other studies that have psychosocial interventions for managing common mental disorders such as depression but also internalized HIV stigma. We are currently piloting a Tendai intervention for managing depression among HIV positive women in Blantyre. We have also developed tools for providing information on treatment as prevention in the hope that to normalize this knowledge within our contexts and reduce HIV related self-stigma.

6. PLOS authors have the option to publish the peer review history of their article (what does this mean?). If published, this will include your full peer review and any attached files.

Do you want your identity to be public for this peer review? For information about this choice, including consent withdrawal, please see our Privacy Policy.

Reviewer #1: No

Reviewer #2: Yes: Estefany Castillo Ravanal

---

## [Editor Report · Decision Letter 1]

27 Mar 2023

Lived experiences of people living with HIV - A qualitative exploration on the manifestation, drivers and effects of internalised HIV stigma within the Malawian context.

PONE-D-22-27597R1

Dear Dr. Kumwenda,

We’re pleased to inform you that your manuscript has been judged scientifically suitable for publication and will be formally accepted for publication once it meets all outstanding technical requirements.

Kind regards,

Nelsensius Klau Fauk, S.Fil., M., MHID, MSc, PhD

Academic Editor

PLOS ONE

Additional Editor Comments (optional):

The authors have carefully addressed the comments from reviewers and improved the quality of the manuscript. It has fullfiled the criteria for publication.
---

## [Editor Report · Acceptance letter]

19 Apr 2023

PONE-D-22-27597R1 

Lived experiences of people living with HIV - A qualitative exploration on the manifestation, drivers, and effects of internalized HIV stigma within the Malawian context. 

Dear Dr. Kumwenda:

I'm pleased to inform you that your manuscript has been deemed suitable for publication in PLOS ONE. Congratulations! Your manuscript is now with our production department. 

Kind regards, 

on behalf of

Dr. Nelsensius Klau Fauk 

Academic Editor

PLOS ONE